

# Study on Flood Control Safety Evaluation Based on Composite Risk Model

Yingying Lan[1], Faliang Gui[1], Dongnan Luo[1], Youqin Zou[2], and Hua Bai[1]

[1]Academy of Hydraulic and Ecology Engineering, Nanchang Institute of Technology, Nanchang, 330099, Jiangxi,China
[2]School of Resources, Environmental and Chemical Engineering, Nanchang University, Nanchang 330031, Jiangxi, China

**Correspondence:** Faliang Gui (falianggui@sina.com)

**Abstract.** The process of dam design up to the management of dam operation involves many uncertain factors, such as hydrologic, hydraulic and flood control factors, which cause risks for the flood control safety of dams. This study presents an integrated probabilistic framework that combines Monte Carlo Simulation and a flood control risk model. Results show that the highest flood level of 1000-year return periods of the Zhelin Reservoir exceeds the designed flood level. However, the overtopping risk probability is small because super safe elevation is considered in the crest elevation design of the earth dam. Sensitivity analysis indicates that flood peak flow, line type hydrological factors and flood control level are more sensitive than hydraulic factors. When many factors are considered, the comprehensive risk rate is small because of the positive and negative effects of these factors. Numerical experiments indicate that hydrology and flood control level influence the estimated maximum water level more than hydraulics does. Because of these uncertain factors, it is necessary to consider super safe elevation in dam planning and design. And pay attention to the sensitive factor of flood control level in reservoir management and operation.

**Keywords**: Risk assessment; Flood control safety; Overtopping probability; Monte Carlo simulation

## 1 Introduction

Floods are regard as one of the most harmful natural disasters (Pinter 2005). In the United States, floods accounted for 63.7%of all disasters during 1953-2010 (Michel-Kerjan and Kunreuther 2011). China experienced 50 major floods over 1900-2010, which affected more than 160 million people, killed 2.5 million people and economic lost $ 16.8 million (Liu et al. 2012).

In recent years, frequent extreme weather events leading to flood disasters increased day by day have become the focus of attention of governments and academia. A lot of organizations and governments have carried out a series of research projects to enhance human disaster prevention capacity and reduce disaster risk since 2000, such as integrated risk management project of flood disaster (Shi et al. 2005), migration project in relation to global environmental change (Black et al. 2011), global risk information platform program (Yin and Xu 2012) and Asia flood network project (Quan 2014). Additionally, China has made floods and other natural disaster risk identification, assessment and monitoring as national research focus (Liu et al. 2012). The flood risk research is an important and effective non-engineering measures for flood control and disaster mitigation (Lv 2015).Assessment on flood risk has been paid attention to and widely studied since 1950s (Richards 1955).



25    The discharge capacity of flood control structures is designed according to the design flood of a specific return period. In practice, the design flood is usually estimated by the frequency analysis using observed runoff data. However, in areas where there is no runoff data or insufficient runoff data, the design flood is generally estimated by adopting a design rainstorm data of a specific return period through the hydrological and hydraulic routing. Using the design flood, the various characteristic water levels from the hydraulic model can be obtained for determining the scale or size of flood control hydraulic structures,

30    such as the crest elevation of dam. However, many uncertainties in the estimation of floods are caused due to imperfect of calculation procedure, incomplete data and randomness of flood (Smith and Ward 1998). This may lead to the actual failure probability of the flood exceeding the design value resulting in potential destruction of the flood control structure. Therefore, it is necessary to carry out flood risk analysis for the hydraulic structures. Tung (1985) evaluated the reliability of hydraulic structures using a model comprehensively consider the hydrological and hydraulic uncertainties. Lee and Mays (1986) analyzed

35    the failure probability of reservoir flood control due to the hydraulic uncertainty factors roughness coefficient of Manning's equation. Apel et al. (2007) proposed dynamic probabilistic model was applied to explore the influence of dike breaches on flood frequency distribution along rivers. The proposed method based on actual flooding data used Monte Carlo framework to simulate the whole flood process and quantify the flood risk, rather than hypothetical scenarios.

    The aforementioned research primarily focuses on the influence of single factor (hydrological and hydraulic uncertainties)

40    for the risk analysis of hydraulic structures. The risk analysis model proposed herein not only takes into account single factors (hydrological, hydraulic, and flood control level) but also composited these uncertainties factors to explore the reliability of the flood control structures of reservoir dams. Since reservoirs and levees are widely used to protect watershed areas from being flooded, this study focuses on a risk analysis for the flood-control ability of reservoir dams in order to calculate the failure probability of the water level exceeding the dam crest elevation. This study evaluates the risk of the maximum water

45    level exceeding the dam crest elevation of the reservoir using the advanced Monte Carlo Simulation. The model requires a corresponding functional relationship between the maximum water level and the hydrological, hydraulic, and flood control level factors, which must be established using integrated probabilistic framework that combines Monte Carlo Simulation and a flood control risk model. This study analyzes the sensitivity of the hydrology, hydraulics, and flood control level uncertainty factors as related to the flood-control capacity of dams along the Xiuhe River, and evaluates the performance of the typical

50    flood and their impact on the flood-control ability of the dam system using the proposed risk analysis model.

## 2   Materials and data

### 2.1   Study area

The Zhelin Reservoir is located in the central part of Jiujiang City, Jiangxi Province, China (Figure 1) (Lan 2014). The study area has a subtropical monsoon with mild and humid climate, four distinctive seasons and abundant rainfall. The Zhelin

55    Reservoir was beginning to build at the end of 1958. But construction on the reservoir was suspended in 1962, and reworked in 1970. Finally, the earth dam filling was basically completed in 1971. The dam length is 590.7 m, the dam height is 62 m, and the rain collection area is 9340 $km^2$. The Zhelin Reservoir is the largest earth dam reservoir in China, with a total





capacity of $7.9 \times 10^9 m^3$, flood control capacity of $3.2 \times 10^9 m^3$, and irrigation storage of $3.44 \times 10^9 m^3$. It is a large water conservancy and hydropower project with comprehensive benefits of flood control, irrigation, shipping and development of

60 aquatic products. The design flood for the flood-control hydraulic structures in the Zhelin Reservoir is 1‰(1000-year return periods, design peak flow of 18250 $m^3$/s), and the standard of checking is 0.1‰(10000-year return periods, design peak flow of 22900 $m^3$/s). On the basis of these standards, the designed flood level is 70.13 m, checking flood level is 73.01 m, and crest elevation is 73.5 m. The design flood level is calculated on the basis of flow data and rainstorm data of 1954-1958 and the data of Historical Extraordinary Flood. During this period, the reservoir construction is in its early stage, and the length

65 of the hydrological data was short. Dams that were designed and built decades ago may not meet current design standards that reflect our improved knowledge of extreme flood events (Byungil Kim 2017). Although dams may decrease the frequency of flooding, they may exacerbate the hazards of flooding (NRC 2012). Therefore, the scale of The Zhelin Reservoir is large, and its safety is particularly important. Whether the standard of flood control hydraulic structures in the reservoir reaches the design requirements, and whether a certain risk of flood control safety exists in the reservoir should be analyzed and evaluated.

70 ## 2.2   Data sources

The raw data source utilized in this research includes the statistical data of flood peak flow (1901, 1931 and 1953-2010 years) from the hydrologic station (Figure 2), type and size of spillway structure and topographic map of the research area. On the basis of the flood data, two typical floods are selected. The typical floods are 1954 and 1998 respectively (Figure 3). Moreover, supported by hydrologic system in the research area, the present study counted the number of floods, recorded the peak flow

of each floods and then established the database for flood analysis.

## 3   Risk analysis model

The Monte Carlo method is used in this paper, that is, the random simulation method. A large number of statistical experiments are carried out according to a suitable probability model, and the probability distribution of random variables. Based on the experimental results, the solution of the stochastic problem can be gotten by analyzing and inferring. Compared with other

methods, the Monte Carlo method has the following advantages: (1)The speed of convergence is independent of the model dimension; (2)The solving is impacted weakly by the stochastic problem; (3)The structure of the program is simple and the result of calculation is more accurate. The key of the Monte Carlo method to calculate risk is to generate random numbers of the known random variables first. Then, a large number of simulation operations are carried out by computer, and the risk is estimated by simulation results.

## 3.1   Generation of random numbers

It is necessary to generate random numbers of the random variables with known distribution types using Monte Carlo method to simulate failure probability. Then, the distribution type is transformed into a random parameter according to the distribution



function of the random variable. Taking the random number of peak flow as an example, the specific methods and steps are as follows:

(1)The peak flow of natural river obeys the distribution of Pearson type III (P-III) (Dong and Wang 2003). The peak discharge series upstream of the reservoir with the distribution of P-III (Formula 1) is fitted according to the available data. The mean value, variance, variation coefficient and skewness coefficient of the statistical parameters are calculated.

$$f(x) = \frac{\beta^{\alpha}}{\Gamma(\alpha)}(x - \alpha_0)^{\alpha-1}e^{-\beta(x-\alpha_0)} \tag{1}$$

where,

$$\alpha = \frac{4}{C_s^2} \qquad \alpha_0 = \bar{x}(1 - \frac{2C_v}{C_s}) \qquad \beta = \frac{2}{\bar{x}C_vC_s} \tag{2}$$

And $\Gamma(\alpha)$ represents $\Gamma$distribution,$\bar{x}$ represents mean value, $C_v$ represents coefficient of variation, $C_s$ represents coefficient of skewness.

(2)The methods of generating random numbers include multiplicative congruence method and mixed residual method. Especially, multiplicative congruence method is widely used because of its' excellent statistical characteristics. The pseudorandom numbers of (0, 1) distributed in this paper are obtained by multiplicative congruence method. The homogeneity test and independence test have been carried out, and the results are satisfactory.

(3)The $Q_m$ random series of flood peak will be obtained by the transformation method. According to the typical flood, the flood process series can be calculated by the same frequency amplification method, and then the random flood data can be obtained.

## 3.2 Risk analysis of dam flood control

Dam flood control risk means that the reservoir results in a failure probability of dam accident because it does not defeat against flood under the height of the dam top and the scale of the flood discharge building.

Floods in China are usually divided into rainstorm floods, flash flood, debris flows, ice floods, melting snow floods, storm surge floods, dam bursting flood and overbank flood according to the formation mechanism of flood disaster. Among them, rainstorm floods occur most frequently, with the largest magnitude and the widest impact. Dam bursting flood is caused by sudden burst of dam or other water retaining structures. Overbank flood refers to the continuous rise of river water leading to flooding of riverbanks. In the past flood risk research, the definition of the risk is a failure probability, such as the probability of water level rising over the dam crest, which is named as the classical risk. In order to study the quantitative estimation formula of dam flood control safety risk the type of risk must be identified in advance. The classical risk is adopted in this study. Wu and Yang (2011) indicated that the risk for reservoir flood control analysis can be expressed as follows:

$$P_{risk} = P_r[L > R] \tag{3}$$

where $P_r(\cdot)$ refers to overtopping probability that system loading (L) is greater than the resistance (R). For the flood control ability of the dam, L and R can be, respectively, the maximum water level and the crown elevation of the dam, and Eq. 3 can





be rewritten as,

$$P_{risk} = P_r[Z_{max} > H_{dam}] = \frac{S}{M} \times 100\% \tag{4}$$

where $Z_{max}$ represents the maximum water level, and $H_{dam}$ represents the height of the dam crown. M refers to total number of random tests, S refers to times of $Z_{max} > H_{dam}$

### 3.3 Generation of uncertainty factors

Theoretically, the hydrological, hydraulic, and flood control level factors have different physical and statistical characteristics.
As a result, this study generates the uncertainty factors using the Monte Carlo simulation based on their physical and statistical properties. The generation of the uncertainty factors is described below.

#### 3.3.1 Hydrological uncertainty factors

Flood is a very complex dynamic stochastic process, usually described by flood characteristics such as flood peak, flood volume and flood hydrograph. Flood peak and flood hydrograph are the mainly risk for flood control according to analysis
of hydrological data. Based on the existing flood data analysis, two typical flood hydrographs in 1954 and 1998 are selected (Figure 3). Then, according to the simulation method mentioned above, flood sequence can be obtained and flood routing can be done. And then, the rate of flood risk ($P_r$) can be obtained by formula 3. The hydrological frequency calculation is based on the data of 1953-2010 years and the extraordinary flood of 1901 and 1931.The frequency calculation results are shown in Figure 4 and Table 1. Because of the different data used, the result of this calculation is different from that of past design results.
The past design flood peak and check flood peak were 18250 $m^3$/s and 22900 $m^3$/s respectively. And results of according to 1953-2010 data are 18621 $m^3$/s and 23643 $m^3$/s respectively. The calculation results show that the design flood peak and check flood peak are increased by 2.03% and 3.24% respectively compared with the past. It can be inferred that there may be some potential risk in the Zhelin Reservoir according to the design standard. So this paper then focuses on the flood risk of the Zhelin Reservoir.

#### 3.3.2 Hydraulic uncertainty factors

The main hydraulic uncertainties factors of flood risk are form and size of spillway structure, and uncertainty of discharge coefficient because of coefficients of hydraulic structures (weir coefficient and tube flow coefficient). Through the statistical analysis of many scholars it is thought that most of the errors caused by these uncertainties are normal distribution (Yang 1999). Based on the hydraulic model test results, the study assumes that the total discharge capacity of the flood discharge facilities
varies from 95% to 105% of the design discharge capacity. A correction coefficient of flow is introduced, $\lambda_2$ ($\lambda_2 = 0.95 \sim 1.05$, $\lambda_2$ obeys normal distribution). That can be expressed as follows:

$$Q_{dis.} = \lambda_2 Q_{des.} = (0.95 \sim 1.05)Q_{des.} \tag{5}$$

where, $Q_{dis.}$ represents the actual discharge of reservoir. $Q_{des.}$ represents the design outflow of reservoir.



### 3.3.3 Flood control level uncertainty factors

The flood control level is the highest allowing water level of a reservoir during flood season, and it also is the starting stage of a reservoir for flood control operation during flood season. For a reservoir in operation, the starting stage of flood regulation level is not exactly the same as the flood control level. It takes a certain distribution within a certain interval. That is $Z_0 = (Z_{con.})$.The distribution function of $Z_0$ can be obtained through statistical analysis of actual operational data. According to the actual operation data, the flood control level of the Zhelin Reservoir is 64 m, the starting stage of flood control level is usually

between 63.36 and 64.64 m. A correction coefficient $\lambda 3$ is introduced ($\lambda_3 = 0.99\sim 1.01$, $\lambda_3$ obeys normal distribution). The $Z_0$ can be expressed as follows:

$$Z_0 = \lambda_3 Z_{con.} = (0.99 \sim 1.01)Z_{con.} \tag{6}$$

where, $Z_0$ represents the actual starting stage of flood control level. $Z_{con.}$ represents the design flood control level.

### 3.3.4 Calculation of comprehensive risk

In this study, the comprehensive risk rate calculation is based on the comprehensive consideration of three factors namely hydrology, hydraulics and flood control level. And the risk of the random combination of the three factors is calculated based on the flood regulation process.

Each flood peak generated randomly by P-III distribution is expressed with $\lambda_1$ (the sampling times of $\lambda_1$ are recorded as $M_1$), and the flood process series can be gotten by the same frequency amplification method based on the typical floods. According

to the normal distribution, the discharge curve is generated expressed with $\lambda_2$ (the sampling times of $\lambda_2$ are recorded as $M_2$), and flood control level expressed with $\lambda_3$ (the sampling times of $\lambda_3$ are recorded as $M_3$). The maximum water level of reservoir is expressed with $Z_{max}$ that was calculated based on flood routing. M ($M=M_1 \times M_2 \times M_3$) maximum water levels are obtained for each typical flood by flood routing. Statistics of the times of $Z_{max} > Z_{check}$ in M that is expressed with S. Then, the comprehensive risk rate is calculated according to Formula 3. The computing framework is shown in Figure 5. $[Q_{min}, Q_{max}]$

is the flood peak sampling interval, $Q_{min}$ represents the smallest peak in history, the design flood peak flow is adopted as $Q_{max}$, $Q_d$ (I) is the typical flood data, $Z_{check}$ represents the check flood water level, $Z_{con.}$ represents the flood control level.

## 4 Results and discussion

All factors have different degrees of effects on the flood risk of dams. Such effects can be determined by calculating and analyzing the influence of (1) hydrology, (2) hydraulics and (3) flood control level on the flood risk of dams. The influence

statistics of single factor, the double factor and the comprehensive factor are shown in Table 2. The design flood level is 70.13 m, check flood level is 73.01 m and the dam top elevation is 73.5 m.





### 4.1 Statistical analysis of uncertainty factors

Take the design flood level as a reference. The calculation results show that the factors will put the flood control safety of the Zhelin Reservoir at risk, but the risk value is not large. When a large number of factors are considered, the risk value is small in the portfolio risk analysis because of the positive and negative effects of these factors. Moreover, typical flood is different, and the calculated result of the risk value is different.

The results based on the typical flood of 1954 are expressed as follows. The highest water level in front of the dam is 70.187 m, and the risk rate is 2%, when only random hydrological factors are considered. The maximum water levels and risk rates are 70.226 or 70.2274 m and 1.18% or 1.16% respectively, when hydraulic or starting water level factors are considered. The maximum water level and risk rate are 70.345 m and 0.97%, respectively, when three factors are simultaneously considered. Considering one and two factors have no remarkable effect. The maximum water level increases less than 0.1 m, and the risk value is less than 2%. The three factors should be synthetically affected in practice. On the basis of the calculation, the maximum water level increases by 0.215 m and the risk rate will be 0.97%.

Meanwhile, the results based on the typical flood of 1998 are expressed as follows: The highest water level in front of the dam is 70.489 m, and the risk rate is 5% when only random hydrological factors are considered. The maximum water level and risk rate are 70.502 m and 5.48% and 5.68%, respectively, when hydraulic and starting water level factors are considered. The maximum water level and risk rate are 70.518 m and 5.17%, respectively, when three factors are simultaneously considered. The influence of single or double factors for the typical flood of 1998 is greater than that for the typical flood of 1954. The maximum water level increases by more than 0.35 m, and the risk value is more than 5%. As previously mentioned, the three factors should be synthetically affected in practice. On the basis of the calculation, the maximum water increases by 0.388 m and the risk rate increases by 5.17%.

### 4.2 Discussion

The hydrological data of the Zhelin Reservoir after its construction are used to evaluate whether the reservoir safety meets the design criteria. The result shows that a certain degree of risk is observed in the dam. The length of hydrological data is limited because of the reservoir construction was in its early stage during the study period. The actual designed value of the reservoir does not meet the design standards. The length of hydrological data is important in design as it is used to determine the representativeness and reliability of the results of hydrological calculations. In addition, a certain degree of risk is observed in the reservoir by considering the influence of hydraulics and starting water level. The safety of water conservancy projects can be ensured when these random factors are considered during dam design.

According to the risk calculation results, the maximum water level and risk values of the 1998 flood are higher than those of the 1954 flood. Different typical floods have a remarkable impact on the calculation results, and floods with double peaks are more harmful to reservoir dams than that of single peak. Peak shape, which does not favor reservoir flood control, should be consider to ensure dam safety.



In this study, a sensitivity analysis of three factors namely, hydrology, hydraulics and flood control level is conducted. For the sensitivity analysis, one factor changes by ($\pm 5\%$), and other factors remain unchanged to calculate the highest water level and risk value. The calculation results show that the hydrological factors have the most remarkable impact on risk value (risk value increases by 3.7%) and that the flood control level has the most remarkable influence on the highest water level (highest water level increases by 0.61m). Meanwhile, the hydraulic factor sensitivity is the lowest, and its influence is irrelevant (Table 3). The reliability of hydrological data has remarkable influence on the results of hydrological design. Thus, the design of reservoir dams should focus on the rational analysis of hydrological calculation results. For reservoir operation, the influence of the accuracy of the initial water level on reservoir flood control safety can't be ignored.

The Zhelin Reservoir is a large earth-rock dam reservoir, and its safety is important. According to the risk analysis, the highest water level exceeds the design standard 0.388 m, and the risk rate of the reservoir is 5.17%. However, the occurrence probability is small because of design flood with a 1000-year return period. Moreover, the safe super elevation $h_{saf.}$=1.5 m is considered in dam design.

$$H_{dam.} = Z_{des.} + H_{win.} + h_{saf.} \tag{7}$$

where, $H_{dam.}$ represents the height of the dam crest, $Z_{des.}$ represents the design flood water level, $H_{win.}$ represents the design wind and wave climbing height, and $h_{saf.}$ represents the super safe elevation (based on design specifications).

Although the highest water level exceeds the design flood level by 0.388 m, it is lower than the safe super elevation, and the dam overtopping probability is small. The reservoir dam does not have potential safety risks. It is also shown that the safe super elevation value should be considered in the design of reservoir dams.

## 5 Conclusions

The evaluated result shows that the designed water level of the Zhelin Reservoir does not meet the design standard. Nevertheless, the safe super elevation value is considered in the design of dam crest elevation, and the final result of the evaluation is that the reservoir is safe. It is of great significance for a reservoir to consider the safe super elevation value in design. The influence of hydrology and flood control level factors on flood risk is more sensitive than that of hydraulic factors. The stochastic factors of hydrology and flood control level should be analyzed in design a reservoir. The design results are more safe and reliable by using Historical Extraordinary Flood and disadvantageous reservoir flood as basis of calculation.

*Acknowledgements.* The research was supported by Education Department of Jiangxi Province, China (Grant No. GJJ180922) and Water Resources Department of Jiangxi Province, China (Grant No. KT201726) and the National Natural Science Foundation of China (Grant No. 31660234, 61663029, 51669014).



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

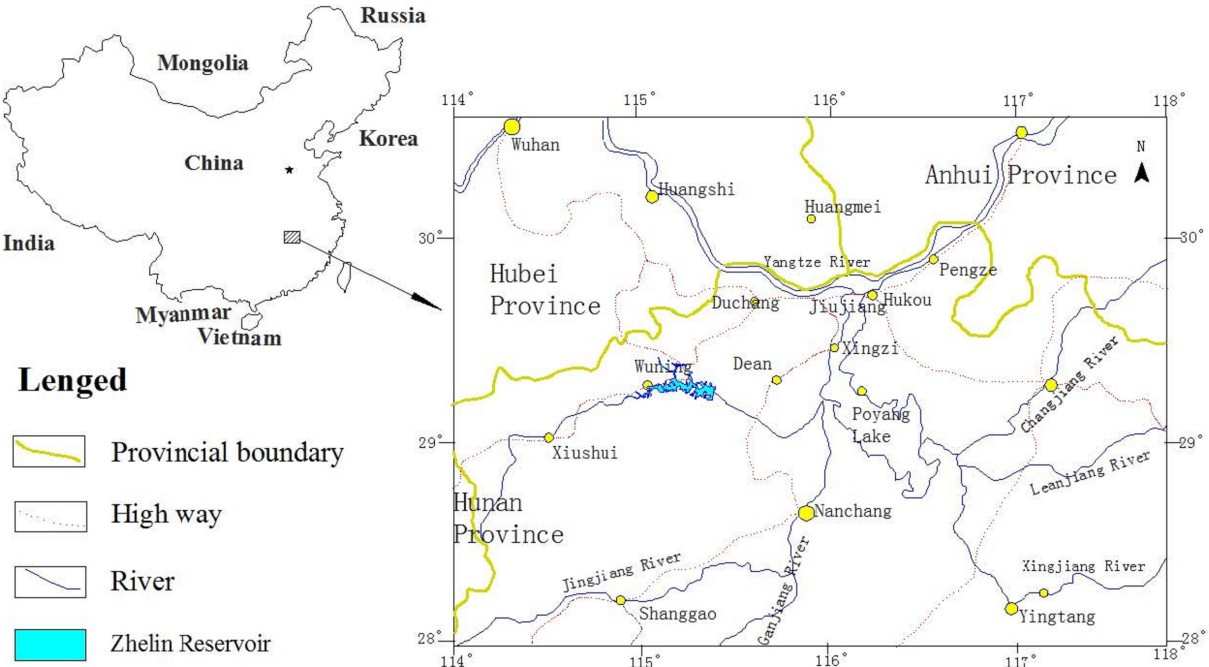

**Figure 1.** Location map of study area

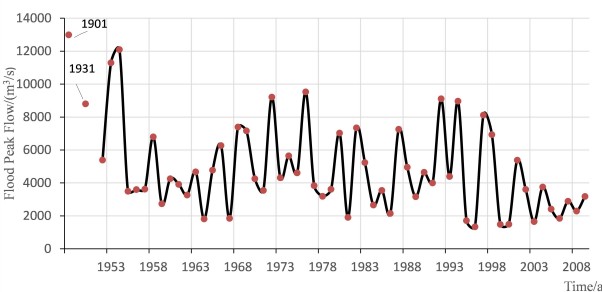

**Figure 2.** Flood peak flow graph of 1953-2010

**Table 1.** Comparison of hydrological frequency calculation results of the Zhelin Reservoir

| Peak flow frequency | Return periods | Past design results | Results of according to 1953-2010 data |
|---|---|---|---|
| 1‰ | 1000-year | 18250 $m^3$/s | 18621 $m^3$/s |
| 0.1‰ | 10000-year | 22900 $m^3$/s | 23643 $m^3$/s |

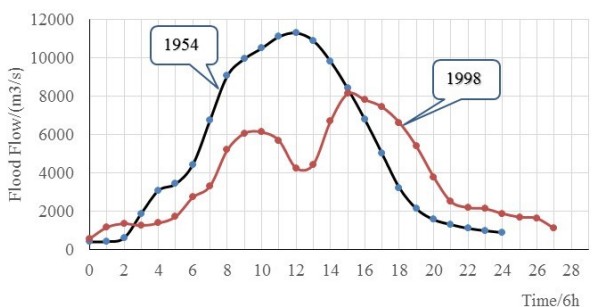

**Figure 3.** Typical flood hygrograph (1954 and 1998)

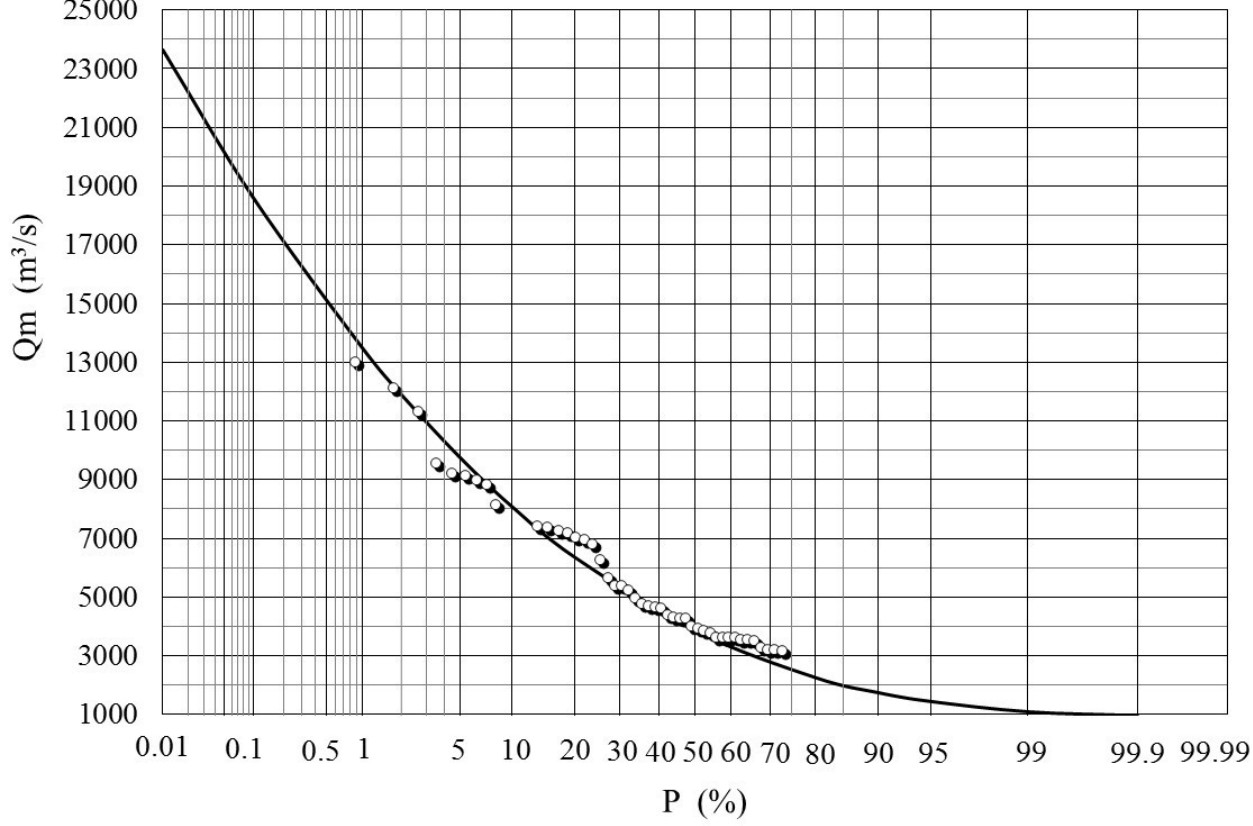

**Figure 4.** Flood peak frequency curve

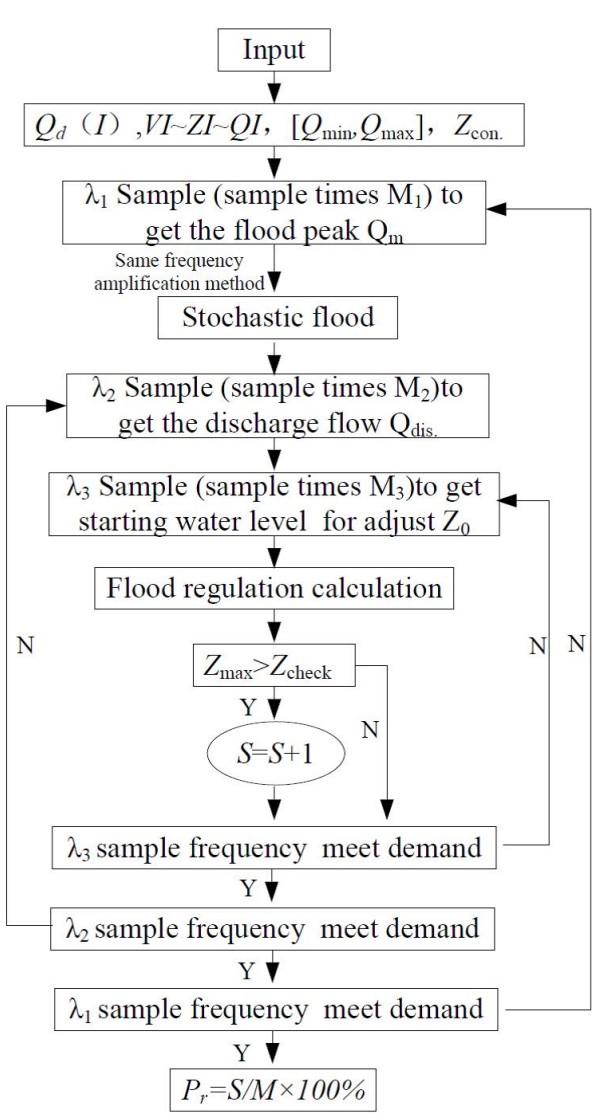

**Figure 5.** Frame diagram of risk calculation




**Table 2.** Statistical table of flood control risk of the Zhelin Reservoir

| Type | Influence factor | Typical flood of 1954 | | | Typical flood of 1998 | | |
|---|---|---|---|---|---|---|---|
| | | (1) | (2) | (3) | (4) | (5) | (6) |
| Sf | A | 70.187 | 2 | 0.057 | 70.489 | 5.00 | 0.359 |
| Df | A&B | 70.226 | 1.18 | 0.096 | 70.502 | 5.48 | 0.372 |
| | A&C | 70.227 | 1.16 | 0.097 | 70.502 | 5.68 | 0.372 |
| Cf | A&B&C | 70.345 | 0.97 | 0.215 | 70.518 | 5.17 | 0.388 |

Sf represents Single factor, Df represents double factor, Cf represents comprehensive factors; A represents hydrologic factor, B represents hydraulic factor, C represents flood control level factor; (1) The highest flood level (m); (2) Percentage of exceeding design flood level ( $Zd_{es.} = 70.13m$)(%);(3) Maximum of exceeding the design flood level (m); (4) The highest flood level (m), (5) Percentage of exceeding design flood level ( $Zd_{es.} = 70.13m$)(%) ; (6) Maximum of exceeding the design flood level (m)

**Table 3.** Sensitivity analysis of the three factors

| Type | (1) | (2) | (3) | (4) |
|---|---|---|---|---|
| Comprehensive factors | 70.345 | 0.97 | 0.000 | 0 |
| hydrologic factors ±5% | 70.532 | 4.67 | 0.187 | 3.7 |
| hydraulic factors ±5% | 70.346 | 1.07 | 0.001 | 0.1 |
| flood control level ±5% | 70.955 | 3.68 | 0.610 | 2.7 |

(1) The highest water level of each type(m); (2) Percentage of exceeding design water level ($Z_{des.} = 70.13m$)(%); (3) Variation of maximum water level relative to comprehensive factors. (m); (4) Variation of percentage of exceeding design water level relative to the comprehensive factors.(%)