# Peer review of "Study on Flood Control Safety Evaluation Based on Composite Risk Model"

_Natural Hazards and Earth System Sciences, 2019_

## Referee Comment (RC1) · Anonymous Referee #1 · 27 Aug 2019

This paper in its present form is not suitable for publication for scientific reasons as well as reasons of poor English.

There is nothing new or novel about the content of the paper to the extent that it reads as if it is a case of a "simulation model looking for an application" that can then be published in a scientific journal.

From the perspective of dams engineering the problem to be solved is totally underspecified and over simplified as the dam and reservoir system is only described in geographic terms with no meaningful engineering specification.

If the subject matter of this manuscript is to be considered for publication then it will require complete reformulation with proper explanation. The deficiencies in the explanation are highlighted in the text.

The lack of any engineering specification beyond the hydrology is a fundamental deficiency.

Please also note the supplement to this comment:
https://www.nat-hazards-earth-syst-sci-discuss.net/nhess-2019-236/nhess-2019-236-RC1-supplement.pdf

**Supplement:**

This paper in its present form is not suitable for publication for scientific reasons as well as reasons of poor English.

There is nothing new or novel about the content of the paper to the extent that it reads as if it is a case of a *"simulation model looking for an application"* that can then be published in a scientific journal.

From the perspective of dams engineering the problem to be solved is totally underspecified and over simplified as the dam and reservoir system is only described in geographic terms with no meaningful engineering specification.

If the subject matter of this manuscript is to be considered for publication then it will require complete reformulation with proper explanation. The deficiencies in the explanation are highlighted in the text. The lack of any engineering specification beyond the hydrology is a fundamental deficiency.

[revised manuscript text omitted]

Sf represents Single factor, Df represents double factor, Cf represents comprehensive factors; A represents hydrologic factor, B represents hydraulic factor, C represents flood control level factor; (1) The highest flood level (m); (2) Percentage of exceeding design flood level ( $Zd_{es.} = 70.13m$)(%);(3) Maximum of exceeding the design flood level (m); (4) The highest flood level (m), (5) Percentage of exceeding design flood level ( $Zd_{es.} = 70.13m$)(%) ; (6) Maximum of exceeding the design flood level (m)

**Table 3.** Sensitivity analysis of the three factors

| Type | (1) | (2) | (3) | (4) |
|------|-----|-----|-----|-----|
| Comprehensive factors | 70.345 | 0.97 | 0.000 | 0 |
| hydrologic factors ±5% | 70.532 | 4.67 | 0.187 | 3.7 |
| hydraulic factors ±5% | 70.346 | 1.07 | 0.001 | 0.1 |
| flood control level ±5% | 70.955 | 3.68 | 0.610 | 2.7 |

(1) The highest water level of each type(m); (2) Percentage of exceeding design water level ($Z_{des.} = 70.13m$)(%); (3) Variation of maximum water level relative to comprehensive factors. (m); (4) Variation of percentage of exceeding design water level relative to the comprehensive factors.(%)

---

## Author Comment (AC1) · 1 Sep 2019

We thank the reviewers for their careful read and thoughtful comments on previous draft and hope you will not hesitate to point out any grammatical or expressive errors in the manuscript. We will ask our experienced English friends to help us revise the manuscript. The Chinese dam elevation design standard was considered to add in the manuscript and making corresponding analysis and discussion.

---

## Referee Comment (RC2) · Anonymous Referee #2 · 8 Sep 2019

The study adopts a numerical modelling approach to investigate flood risks for Zhelin Reservoir, China. In particular, the study uses a probabilistic modelling framework that relies on Monte Carlo simulation and a flood risk model. The topic is certainly relevant to the audience of NHESS, but I do have some major reservations, which are outlined below.

Novelty: the Introduction fails to frame the study within the existing literature on flood risk management. Specifically, the Introduction does not identify significant research gaps and explain how this study addresses them. My understanding is that the study does not advance the state of the art on flood risk management, but simply applies existing techniques to a case study.

Implementation: while the choice of a Monte Carlo framework is reasonable, its imple-

mentation is totally unclear. The study lacks the most basic elements of any numerical study, such as 1) a detailed description of the adopted methods, 2) an explanation of the modelling assumptions (along with the corresponding limitations), and 3) a thorough validation.

Presentation: the quality of the presentation is very poor. Apart from the problems mentioned above, there are several unclear statements, typos, and grammatical errors.

Considering the breadth and depth of all these issues, I believe the manuscript should be rejected.

---

## Author Comment (AC2) · 22 Sep 2019

We thank the reviewers for their careful read and thoughtful comments on previous draft. We have asked our experienced English friends to help us check our English expressions. At present, the type of random flood distribution is still an empirical method, which has not yet reached the theoretical depth. This study is a comprehensive application of some empirical methods to solve flood risk problems, and no new theoretical methods have been put forward.